# Self-efficacy mediates the relationship between grit and life satisfaction in a sample of employed university students resident in the United Arab Emirates

Ektha Benny ⬤, Zahir Vally ⬤*

Department of Clinical Psychology, United Arab Emirates University, Al Ain, UAE

* zahir.vally@uaeu.ac.ae

## Abstract

Balancing academic and professional responsibilities presents unique psychological challenges for employed university students. Most existing literature on this student population has adopted a deficits-approach. However, identifying the strengths inherent in such individuals could offer valuable avenues for informing strategies to reduce their risk of psychological distress. This study examined whether self-efficacy impacts the relationship between grit and life satisfaction among employed university students in the United Arab Emirates (UAE). A cross-sectional study was employed with data collected from 242 employed university students residing in the UAE. Participants completed a self-report survey that comprised of measurements of grit, self-efficacy and life satisfaction. Grit was significantly and positively associated with both life satisfaction and self-efficacy. Additionally, self-efficacy was found to significantly mediate the relationship between grit and life satisfaction in this sample of participants. This study identified grit as a potential factor that could be harnessed to improve life satisfaction, through the mediating role of self-efficacy. These findings hold important implications for informing strategies to promote positive mental health outcomes among employed university students who possess the strength of grit.

## Introduction

Rising living costs [1], the need to meet often convoluted professional qualification requirements [2] and increased competition in the job market [3] have led to an increasing number of students taking up paid employment alongside their education. In the United Arab Emirates (UAE), face-to-face classes for graduate courses are scheduled in the evening, with the intention of accommodating the needs of students who may be working during the morning hours but wish to attend classes after the working day. Research documenting experiences of employed university students within the Middle Eastern context have been limited, with very few addressing the

**Data availability statement:** All relevant data are within the paper and its Supporting Information files.

**Funding:** The author(s) received no specific funding for this work.

**Competing interests:** All authors declare that they have no competing interests to disclose.

dysfunction among this student population [4]. However, in recent times, the field of positive psychology has been gaining momentum, especially in the Arab world [5], with increasing research focusing on a strengths-based approach instead of a deficit-based approach [6].

There is growing consensus that identifying and capitalizing on strengths can nurture success [7]. Fostering psychological strengths could play a crucial role in preventing the occurrence of mental health difficulties, as it would aid in reducing the susceptibility to psychological distress by building psychological resources to mitigate the effects of any pre-existing risk factors [8]. This study aimed to examine the potential pathways through which employed university students in the UAE – an understudied population in the current context – can leverage their strengths to improve their mental health, thereby promoting a more rewarding educational experience.

## Review of literature

### Employed university students

Existing literature has documented that employed university students are more likely to report difficulties in balancing the simultaneous demands of their academic degree and employment [9–12]. They often tend to graduate later [13], enrol in fewer courses per semester and have lower attendance rates compared to their non-employed peers [1,3,14]. Furthermore, a lack of sufficient time for leisure and social activities can hinder their ability to recuperate [15], which may explain the rising levels of stress [16,17] and burnout [10,12,18]. This is concerning because burnout increases the risk of psychopathology, including suicidal ideation [19,20] and depression [21,22]. Bourchak [13] reported that Middle Eastern male university students who are simultaneously employed are more likely to drop out of their courses of study. This is likely the result of a culturally-oriented view of financial independence for males that appears to precipitate the perception that it is generally unacceptable to be financially dependent on their families beyond the age of 18. Additionally, many students marry at comparatively younger ages, compared to their first-world, Western counterparts, and must support their own families. As a result, they are obliged to work and attend evening classes after long workdays. The resulting physical and mental exhaustion are more likely to affect their ability to focus, which in turn affects their academic performance and well-being [23]. This challenge may be even greater for employed female students who often take on a larger share of childcare and family responsibilities along with their academic and professional commitments [24]. Life satisfaction, which refers to an individual's cognitive evaluation of satisfaction with their life [25] is negatively affected due to the multiple and often competing demands placed on the employed student [26].

Much of the existing research that highlights the challenges that employed students face has adopted a deficit-based approach, one that focuses on the identification of dysfunction and its remediation [6]. However, competing research demonstrates that, despite various challenges, some students who work part-time do not report being dissatisfied with their overall university experience [1]. For example, in a study conducted in Kuwait among female adult students, many of whom were

also employed, participants reported that governmental support, such as funding and paid leave from work, as well as family support, particularly from their spouses, were crucial in allowing them to balance their multiple responsibilities and complete their studies [27]. Several external factors may influence employed students' experience, including whether their employment is related to their field of study [15,28], the design and mode of delivery of the degree [9,29] whether they are employed on a full-time versus part-time basis [1], and the number of hours worked per week [17]. Thus, multiple variables, encompassing both strengths and weaknesses, need to be considered when analysing the experiences of employed university students. Since prior research has primarily focused on the challenges faced by these students, there is a need to explore their strengths in order to develop a nuanced and comprehensive understanding of their experiences.

## Grit

One strengths-based construct that has been extensively studied in academic settings is grit. Grit is defined as "perseverance and passion for long-term goals" and further refers to an individual's ability to sustain their efforts and remain committed to achieving a goal over several years [30]. Research has linked grit to various positive academic outcomes, including better performance in National Spelling contests [30], higher grade point averages [31], and greater academic resilience [32]. Investigating grit in the context of employed university students is particularly relevant, as grit has also been identified as a predictor of retention in education programs [30,33,34], and employed university students are known to be at a greater risk for attrition [1,13]. Furthermore, individuals with elevated levels of grit have been shown to be more committed to multiple life domains, including work, family and school [35], which may be especially advantageous for employed university students as they manage multiple demands simultaneously. Given these findings, grit may represent a valuable non-cognitive personality trait for employed university students, as it could contribute to enhancing life satisfaction despite the presence of immense life demands. Previous studies have documented a positive association between grit and life satisfaction [36–40] among college students. Moreover, factors such as depression [37], academic adaptation [38] and self-efficacy [41], have been found to mediate the relationship between grit and life satisfaction.

## Self-efficacy

Self-efficacy, defined as "an individual's belief in their capacity to implement tasks and attain self-stated goals" [42], has been linked to a lower incidence of deleterious psychological outcomes, including depression [43], anxiety [44] and post-traumatic stress symptoms [45]. Furthermore, it is associated with several beneficial psychological outcomes, such as improved stress management [46] and a higher sense of general well-being and self-esteem [47]. In the same vein, it has positive effects on the tendency to adopt a healthy lifestyle, as well as faster recovery from surgery or illness [46]. Self-efficacy was positively associated with life satisfaction among young adults [48,49] and was suggested as a pathway for improving life satisfaction in this population [50]. Additionally, self-efficacy was also found to be positively associated with grit [41,51]. Si-Thu et al. [41] reasoned that for South Korean nurses working in high-stress environments, grit may enhance life satisfaction. They conceptualised grit as a relatively stable, enduring "distal trait" that influences life satisfaction indirectly through the more malleable "proximal trait" of self-efficacy [52], which is more closely linked to behaviour. This suggests a pathway in which personality characteristics exert their effect through intermediate factors – such as self-efficacy – that can be developed or enhanced through interventions.

## Theoretical framework

In an integrative review, Datu et al. [53] proposed the Optimal Performance and Health (OPAH) model of grit, which outlined two main pathways through which grit influences psychological functioning. The first pathway posits behavioural effort, adaptive motivation and cognitive resourcefulness as mechanisms through which grit facilitates achievement. Accordingly, grittier individuals tend to invest sustained effort, possess stronger motivation to succeed, and more effectively utilise cognitive strategies, thereby promoting goal attainment. The second pathway highlights three mechanisms

linking grit to well-being – needs satisfaction, emotion regulation and positive cognitions. However, the construct of positive cognitions proposed in the model, which encompasses positive thoughts about the self, others and the world is conceptually broad and lacks specificity. Moreover, the model does not elaborate on how these positive cognitions manifest in real-life contexts to enhance well-being. Several studies have examined variables closely related to positive cognitions – such as self-efficacy [41], meaning in life [54], self-esteem [55], mindfulness [56] – as mediators in the relationship between grit and well-being. Drawing on this rationale, self-efficacy can be conceptualised as a specific and measurable manifestation of positive cognition, serving as a potential pathway through which grit may impact life satisfaction among employed university students. Although the OPAH model refers broadly to well-being, life satisfaction is considered a key dimension of well-being [25] and is widely used as an indicator [57]. Moreover, unlike other dimensions of well-being, life satisfaction reflects more stable life aspects and is derived from multiple domains of life; such as work, relationships, and health [58]. Thus, it provides a broader and more enduring assessment of an individual's overall well-being. Based on this understanding, the OPAH model can be applied to explain the relationship between grit and life satisfaction in the current study.

As individuals consistently engage and stay committed to achieving long-term goals (e.g., a university degree program that likely spans many years), for grittier students, the accumulating experience enables them to overcome obstacles that emerge during the course of task completion. This promotes acquisition of cognitive and behavioural resources to better cope with stress, which literature suggests is associated with improved self-efficacy [46,59]. Additionally, it is further contended that students who are concurrently employed alongside their studies tend to be mature students, those beyond the young adult years, and purportedly with advancing age and the accumulation of life experiences over time, self-efficacy increases as individuals develop better physical, psychological, cognitive, social and economic resources [60]. This increases their repertoire of coping skills [61], which when drawn upon to navigate challenges, promotes greater life satisfaction [62]. Building on this theoretical and empirical rationale, the present study examines the potential role of self-efficacy in explaining the relationship between grit and life satisfaction among employed university students in the UAE.

### Aims and hypotheses

The current study aimed to determine whether self-efficacy mediates the association between grit and life satisfaction. Accordingly, two hypotheses were proposed; grit would be positively associated with life satisfaction (H1), and self-efficacy would mediate the association between grit and life satisfaction (H2).

## Materials and methods

### Participants and procedure

A cross-sectional study was conducted to examine the aforementioned hypotheses. Participants were required to meet two criteria to be included in the study: [1] currently enrolled in a university program in the UAE and [2] simultaneously employed whilst engaged in study. Data were collected over the Spring, Summer and Fall semesters of the 2024 academic year (data were collected between 12 March and 30 November 2024), using a combination of convenience, snowball and purposive sampling methods. The researcher distributed the study survey via an online survey management program to employed university students, through university email, online networking platforms and by approaching them in public university spaces. Participants were encouraged to share the survey with eligible colleagues.

The survey, when first accessed, presented participants with a consent form. Written informed consent was obtained from all participants. The information provided to participants on the consent form included an explanation of the study's objectives, the potential risks involved in participation, an indication that participation was voluntary and highlighted their right to withdraw from the study at any time without consequence. To ensure confidentiality, no identifying information was collected. Ethical approval for the conduct of this study was obtained from the Social Sciences Research Ethics Committee at the author's institution before data collection commenced (reference number: ERSC_2024_4293).

The ideal sample size was determined using Monte Carlo Power Analysis for Indirect Effects [63], as this study aimed to determine the mediating effects of self-efficacy on the relationship between grit and well-being. The web version of the application was used (https://schoemanna.shinyapps.io/mc_power_med/). The power was set to 0.8 and medium effect sizes were assumed to be 0.6, 0.39 and 0.39 for the relationship between the independent variable and mediator (*a* path), mediator and dependent variable (*b* path), and indirect effect between the independent and dependent variables with the mediator (*c'* path), respectively [51]. Analysis revealed the ideal sample size to be 174. The final sample consisted of 242 participants, which was well above the ideal sample size.

Table 1 provides an overview of the sample's demographic characteristics. Regarding age, nearly half of the participants were in the 18–25 age group (40.1%, n = 97) and the 26–35 age group (42.1%, n = 102). The majority of the participants were of Arab ethnicity (59.5%, n = 144), female (73.6%, n = 178), pursuing a master's degree (70.7%, n = 171), unmarried (54.5%, n = 132), without children (66.9%, n = 162), and employed full-time (as opposed to part-time) whilst engaged in concurrent university study (69%, n = 167).

**Table 1. Percentages of Demographic Variables.**

| Variables | n (%) |
|---|---|
| Age | |
| Below 18 years | 1 (.4%) |
| 18–25 years | 97 (40.1%) |
| 26–35 years | 102 (42.1%) |
| 36–45 years | 39 (16.1%) |
| Above 46 years | 3 (1.2%) |
| Gender | |
| Male | 64 (26.4%) |
| Female | 178 (73.6%) |
| Type of Employment Status | |
| Full-time | 167 (69%) |
| Part-time | 75 (31%) |
| Course of Study | |
| Bachelors | 10 (4.1%) |
| Masters | 171 (70.7%) |
| Diploma | 1 (.4%) |
| Doctorate (PhD) | 60 (24.8%) |
| Marital Status | |
| Married | 110 (45.5%) |
| Unmarried/ Single | 132 (54.5%) |
| If they have children? | |
| Yes | 80 (33.1%) |
| No | 162 (66.9%) |
| Ethnicity | |
| Arab | 144 (59.5%) |
| South Asian | 76 (31.4%) |
| East Asian | 4 (1.7%) |
| Caucasian | 8 (3.3%) |
| African | 10 (4.1%) |

## Assessment instruments

**Self-efficacy.** Self-efficacy was measured using the General Self-Efficacy Scale (GSES). It is a 10-item self-administered questionnaire which measures an individual's general belief in their ability to respond to and control environmental demands [64]. Respondents provide an indication in response to each item as to the extent to which each statement applies to them. Responses are provided using a four-point Likert scale, where "1" indicates "not at all true" to 4 indicating "exactly true". The total score for the GSES is computed as a total for all responses to the ten items. Higher scores are indicative of an elevated sense of self-efficacy. Example items include: "I can always manage to solve difficult problems if I try hard enough", "I can remain calm when facing difficulties because I can rely on my coping abilities", and "when I am confronted with a problem, I can usually find several solutions". Psychometric evaluation of the GSES indicates its reliability and validity. Internal consistency values typically range from .82 to .93. Test-retest reliability over a two-year period was found to be .47 and .63 for men and women respectively. The Cronbach's alpha for the current study was .91, indicating excellent internal consistency.

**Life satisfaction.** The Satisfaction with Life Scale (SWLS) is a 5-item scale that measures "an individual's global cognitive judgments of one's life satisfaction" [65]. Participants indicate their responses on a seven-point Likert Scale where "1" indicates a response of "Strongly disagree" and "7" indicates a response of "Strongly agree". A total score for the SWLS is obtained as a total of all responses to the five items. Higher scores indicate greater subjective life satisfaction. Example items on the SWLS include: "In most ways my life is close to my ideal", "so far I have gotten the important things I want in life", and "If I could live my life over, I would change almost nothing". It has been reported to have good psychometric properties with excellent internal consistency (Cronbach's alpha = .87) and test-retest reliability (α = .82). In the present study, the measure produced excellent internal consistency (α = .89).

**Grit.** Grit was measured using the 12-item version of the Grit Scale (GS) [30]. Responses are provided using a five-point Likert scale that ranges from 1 ("Not much like me") to 5 ("Very much like me"). Item numbers 1, 4, 6, 9 and 12 measure the factor of perseverance of effort (PE) and items 2, 3, 5, 7, 8 and 11 are reverse scored and measure the factor of consistency of interest (CI). The overall grit score is produced by computing the average of the scores across the twelve items with higher scores indicating higher levels of grit. Example items include: "I have overcome setbacks to conquer an important challenge", "I often set a goal but later choose to pursue a different one", and "I have achieved a goal that took years of work". Internal consistency in the initial validation study was high for the overall score (α = .85) as was the case when the two subdimensions were considered (PE (α = .78) and CI (α = .84), respectively) [30]. In the current study, internal consistency was similarly high (α = .84).

## Data analysis plan

Descriptive statistics were calculated for all demographic characteristics as well as the study's primary variables. These are reflected as means and their associated standard deviations, for continuous variables, and as counts and percentages for categorical variables. A correlational matrix was generated as a preliminary investigation of the potential relationships between the primary variables (grit, self-efficacy and life satisfaction). This resulted in Pearson's correlation coefficients with associated significance values. Then, mediation analyses were performed using the Process Macro version 4.2 (www.processmacro.org/index.html) [66]. A simple mediation model was proposed in which self-efficacy was tested as a single potential mediator in the relationship between grit and life satisfaction. A bootstrapping method with 10,000 samples was applied to the mediation analysis, producing bootstrapped confidence intervals (95% CI). A p value of .05 was indicative of statistical significance.

## Results

### Descriptive results

Table 2 shows the descriptive results for the primary variables. Analysis reveals that the grit scores ranged between 1.92 to 4.83. The maximum score that can be obtained on the GSES is 5, and a grit score of 2.5 is indicative of average levels

**Table 2. Mean, standard deviation, maximum and minimum values for primary variables.**

| Variable | Minimum | Maximum | *M* | *SD* |
|---|---|---|---|---|
| GS | 1.92 | 4.83 | 3.77 | 0.62 |
| SWLS | 8 | 35 | 27.70 | 5.79 |
| GSES | 16 | 40 | 33.08 | 5.11 |

Note. N = 242, GS (Grit Scale), SWLS (Satisfaction with Life Scale), GSES (General Self-efficacy Scale)

of grit [30]. In the current study, the mean GSES score for the present sample was 3.77 (*SD* = .62), which is above the average grit score. This indicates that in the current sample, employed university students in the UAE, have moderately high grit. Furthermore, for life satisfaction, the lowest score recorded was 8 and the highest score was 35. The mean SWLS score was 27.70 (*SD* = 5.79). Since scores obtained on the SWLS that range between 25 and 29 are indicative of a "high" level of life satisfaction [25,67], the mean scores in the current sample indicate "high" life satisfaction for employed university students. Lastly, self-efficacy scores ranged from 16 to the maximum value of 40 (m = 33.08, *SD* = 5.11). On the GSES, higher scores are reflective of higher self-efficacy, with the maximum possible score being 40 [64]. Therefore, in the present sample, the mean self-efficacy score indicates the presence of high self-efficacy.

## Correlational results

Pearson's product moment correlation analysis was carried out to determine the nature and extent of the relationship between the primary variables of the study. It is noted that all primary variables were found to be significantly and positively correlated with each other (see Table 3). A moderate positive correlation was found between grit and life satisfaction, ($r = .67$, $p < .001$). On a similar note, self-efficacy was also found to be significantly and moderately correlated with grit ($r = .61$, $p < .001$). However, the relation between self-efficacy and life satisfaction was found to be strongly correlated ($r = .70$, $p < .001$).

## Mediation results

Results of the mediation analysis are presented in Fig 1. The overall model was found to be significant ($F_{(1, 240)}$ = 199.28, $R^2 = .45$, $p < .001$). Path a, which denotes the association between grit and self-efficacy, was significant ($\beta = 4.99$, SE = .42, 95% CI 4.16, 5.82). Path b was also significant, indicating a significant association between self-efficacy and life satisfaction ($\beta = .53$, SE = .06, 95% CI .41, .65). The total effect of grit on life satisfaction, represented by path c, was significant ($\beta = 6.29$, SE = .45, 95% CI 5.41, 7.16). In addition, both the direct effect (path c) as well as the indirect effect (path c') of grit on life satisfaction were significant. Thus, self-efficacy emerged as a significant mediator between total grit and life satisfaction ($\beta = 2.64$, SE = .42, 95% CI 1.84, 3.51).

## Discussion

This study sought to examine the potential mediational role of self-efficacy in the relationship between grit and life satisfaction among employed university students residing in the UAE. Many university students in the UAE are engaged

**Table 3. Correlation analysis of primary variables.**

| | Variables | 1 | 2 | 3 |
|---|---|---|---|---|
| 1. | GS | 1 | – | – |
| 2. | SWLS | .67** | 1 | – |
| 3. | GSES | .61** | .70** | 1 |

Note. N = 242, GS (Grit Scale), SWLS (Satisfaction with Life Scale), GSES (General Self-efficacy Scale), **p<.001 (Two-tailed)

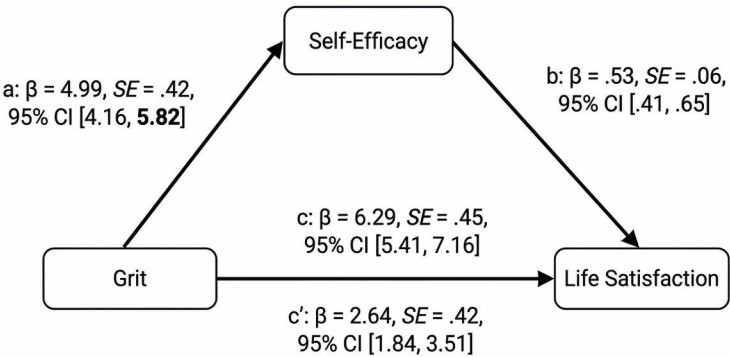

**Fig 1. Results of the mediation analysis indicating a significant effect.**

in paid employment, yet research on their experiences remains scarce. Much of the existing literature on this student population, conducted elsewhere in the world [10,12,16–18], focuses on the detrimental psycho-emotional effects of being an employed student rather than on exploration of their potential strengths, those that might support their well-being and overall life satisfaction. Given that employed students may be at a higher risk of experiencing mental health challenges, fostering psychological strengths could play an essential role in preventing the development of psycho-emotional distress [8]. Therefore, to address this research gap, the current study adopted a strengths-based approach by examining grit, life satisfaction and self-efficacy among employed university students in an understudied part of the world, those resident in the UAE. Results of this study revealed that grit was significantly and moderately positively correlated with life satisfaction, which confirms H1. Furthermore, self-efficacy was determined to significantly mediate the relationship between grit and life satisfaction, thereby confirming H2.

The positive association between grit and life satisfaction observed in this study aligns with findings from previous research [36–40]. A large proportion of participants in this sample were above the age of 25. In line with the findings of Duckworth et al. [30], as individuals grow older, they increasingly realize that success often requires sustained commitment to a specific goal over an extended period. Grittier employed university students – who often tend to be older and mature – are likely to pursue goals over extended periods of time. The longer individuals persist in their efforts to achieve a goal; they refine their ability to acquire effective problem-solving strategies [61]. As a result, they become better equipped to handle challenges efficiently and develop a sense of mastery. Moreover, age and accumulated life experiences, are accompanied by better cognitive, physical, economic and social growth due to biological development and increased exposure to diverse experiences [60], which can expand one's coping skillset. This enhanced capacity to manage life's demands may, in turn, foster stronger beliefs in one's ability to succeed, thereby increasing self-efficacy [46]. Higher self-efficacy has been shown to motivate persistence despite difficulties; and possessing a broad repertoire of skills enables them to overcome obstacles, ultimately contributing to greater life satisfaction [62]. Thus, employed university students who have higher self-efficacy are better able to stay committed to their goals, and efficiently manage multiple responsibilities [35], which may explain the positive link between grit and life satisfaction found in this study [41].

However, several socio-cultural factors may influence how self-efficacy mediates the relationship between grit and life satisfaction [53]. In the current study, the majority of participants were females and belonged to Arab cultures. A significant proportion of students in the sample were also married with children, which poses additional challenges for employed university students [24]. Previous research indicates that Arab female students often rely on family support, particular from spouses, to successfully complete their studies, which may in turn affect their levels of grit, self-efficacy and life satisfaction [27]. Additionally, similar to Kuwaiti female adult learners, Emirati female adult learners also receive governmental

support to fund their education and are granted paid leave from work. Although, the current sample did not entirely consist of Emirati employed university students, this contention may still be plausible, especially since the sample mostly included students from a federal university, where the vast majority of students are Emirati nationals. Such socio-cultural support could affect employed university students' self-efficacy, which may either amplify or attenuate their levels of grit, as well as the extent to which grit impacts life satisfaction.

## Limitations and directions for future research

The findings of the current study highlight a potential pathway that can be leveraged to enhance the influence of grit on life satisfaction among employed university students in the UAE. However, several limitations should be acknowledged. The cross-sectional research design provides valuable insight into the mediating role of self-efficacy but does not explain a cause-and-effect relationship. As indicated by the OPAH model of grit [53], multiple dispositional [51] and contextual factors [37] may moderate the extent to which self-efficacy influences the relationship between grit and life satisfaction among employed university students. Future research employing a longitudinal design – tracking participants across multiple time points throughout their academic journey from admission to graduation – would offer a more comprehensive understanding of how grit, self-efficacy and life satisfaction interact over time. Additionally, examining other potential mediators grounded in positive psychology could yield a more holistic strengths-based understanding of employed university students' experiences. Such variables could perhaps demonstrate a stronger explanatory power than self-efficacy in accounting for the relationship between grit and life satisfaction, which could inform interventions. Moreover, the findings of the current study suggest that future research could adopt randomized controlled designs to experimentally test interventions aimed at enhancing self-efficacy among employed university students with higher levels of grit. Such studies could help determine whether improving self-efficacy enables gritty students to better capitalize on their personal strengths and, in turn, enhance their life satisfaction.

Another limitation of the present study stems from the sampling methodology adopted. The convenience sampling technique relied on online social networking platforms to disseminate the questionnaire with the intention of reaching a broader audience. However, it is likely that most, if not all, participants were drawn from a single university's population. This limitation, coupled with the sample being predominantly female and composed largely of Master-level students, restricts the generalizability of the findings, as they may primarily reflect the experiences of students from this particular institution. Future research could aim to recruit a more diverse and representative sample across the UAE – in terms sex, age, and ethnicity – to enable stratified analyses that provide a more nuanced understanding. Moreover, future studies could also examine the influence of culturally shaped factors in the experiences of employed university students in the UAE. Addressing this research gap could guide the development and implementation of culturally appropriate interventions to support student mental health [13,68].

Finally, the present study did not investigate the presence of psychopathology among employed university students in the UAE, despite existing literature highlighting an increasing prevalence of psychological difficulties among university students in the UAE [4]. Given the limited research focusing specifically on employed university students in the UAE, a lack of understanding regarding the prevalence and nature of psychological difficulties may result in underlying psychological conditions being overlooked. Prospective studies that simultaneously examine both deficits and strengths within this unique student population could offer valuable insights and generate a more comprehensive understanding of their experiences. Such research would serve the dual purpose of addressing psychopathological concerns while leveraging strengths to promote recovery, reduce risk of mental health difficulties and enhance overall well-being.

## Conclusion

The present study adopted a strengths-based approach and demonstrated that the life satisfaction of employed university students in the UAE is associated with their levels of grit and self-efficacy. The study's findings on the mediating role of

self-efficacy highlight a potential mechanism around which interventions could be designed to help employed university students in the UAE to capitalise on their strengths to promote positive mental health outcomes.

## Implications

This research is among the first to examine the experiences of an under-represented student population within a Middle Eastern context, thereby providing a foundation for future studies to expand literature on employed university students. The findings of this study hold significant clinical and policy implications. Policymakers and higher education stakeholders in this region may be encouraged to adopt a strengths-based approach when formulating mental health policies, particularly in university settings. Much of the existing policies and guidance for higher education institutions often emphasize the treatment of distress rather than the proactive promotion of wellness. In contrast, our findings suggest that interventions aimed at fostering grit and enhancing self-efficacy could yield more favourable outcomes [39] – both psychologically by improving life satisfaction [69], and academically by increasing the likelihood of successful course completion [30,33,34].

Drawing on the current findings regarding the mediating role of self-efficacy, institutional policies could focus on improving self-efficacy, so that employed university students may capitalise on their strength of grit to enhance life satisfaction [41]. Currently, employed university students in the UAE attend weekday evening classes after completing work in the morning. Introducing flexible class schedules or blended learning options for courses with a high percentage of employed students could allow students greater control over their schedules. Additionally, short breaks during class could help maintain focus and reduce fatigue. These arrangements would help increase self-efficacy via building mastery and regulating physiological/affective states [42].

Faculty members play a pivotal role in fostering self-efficacy. By being receptive and responsive to students' suggestions and concerns, appropriately using formative assessments, and by providing constructive feedback, faculty can promote students' confidence and adaptability [70]. Additionally, encouraging open discussions and peer support in classrooms can also enhance self-efficacy through vicarious learning [42], as students observe and learn from peers who successfully navigate similar challenges [71]. At the individual level, clinical interventions could help employed university students develop adaptive, problem-focused coping strategies, such as effective time management and seeking faculty assistance [72]. Furthermore, given that approximately half of the employed university students in the sample were married, family and couple therapy is recommended to enhance their well-being in their interpersonal lives [73]. This approach is particularly relevant for employed university students from Middle Eastern cultures, particularly female students, as they often rely on family support to complete their studies [27]. Such therapeutic interventions could build coping skills that strengthen self-efficacy [46,59], enabling students to capitalize on their grit and enhance life satisfaction. Collectively, these findings provide a strong impetus for implementing multi-level strategies to support employed university students in the UAE.

## Supporting information

**S1 File. Complete dataset for the study (Excel Datafile).** Dataset on which the study's analyses are based. (XLSX)

## Author contributions

**Conceptualization:** Ektha Benny.

**Data curation:** Ektha Benny.

**Formal analysis:** Zahir Vally.

**Funding acquisition:** Ektha Benny.

**Investigation:** Ektha Benny, Zahir Vally.

**Methodology:** Ektha Benny, Zahir Vally.

**Software:** Zahir Vally.

**Supervision:** Zahir Vally.

**Writing – review & editing:** Ektha Benny, Zahir Vally.

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
