## [Decision Letter · Decision Letter 0]

28 Aug 2025

Dear Dr. Vally,

Thank you for submitting your manuscript to PLOS ONE. After careful consideration, we feel that it has merit but does not fully meet PLOS ONE’s publication criteria as it currently stands. Therefore, we invite you to submit a revised version of the manuscript that addresses the points raised during the review process.

We look forward to receiving your revised manuscript.

Kind regards,

Mostafa Janebi Enayat, Ph.D.

Academic Editor

PLOS ONE

Journal Requirements:

2. Please ensure that you include a title page within your main document. You should list all authors and all affiliations as per our author instructions and clearly indicate the corresponding author.

Reviewers' comments:

Reviewer's Responses to Questions

**Comments to the Author**

1. Is the manuscript technically sound, and do the data support the conclusions?

Reviewer #1: Yes

Reviewer #2: Yes

2. Has the statistical analysis been performed appropriately and rigorously?

Reviewer #1: No

Reviewer #2: No

3. Have the authors made all data underlying the findings in their manuscript fully available?

Reviewer #1: No

Reviewer #2: Yes

4. Is the manuscript presented in an intelligible fashion and written in standard English?

Reviewer #1: Yes

Reviewer #2: Yes

Reviewer #1: Dear Editor,

Thank you very much for inviting me to review this study. This is an interesting study. The manuscript could benefit revision based on the following comment:

• Abstract: the authors claim to measure ‘well-being’, which is incorrect.

• Abstract: There is no need to refer to Beta and CI values.

• The opening paragraph of the introduction section is too broad.

• The authors are suggested to more clearly explain the research problem and highlight significance of their study in the introduction section.

• The concept of life satisfaction requires further elaboration, as its current introduction lacks sufficient depth.

• The introduction and literature review sections should be clearly separated and numbered.

• I suggest authors avoid discussing measurement issues concerning grit in literature review.

• The authors claim that the “OPAH model proposes that grit enhances life satisfaction via three psychological processes: needs satisfaction, emotional regulation and positive cognitions”. However, I think this is problematic. The OPAH model is about balancing optimal performance with optimal health by maintaining equilibrium between effort and recovery, performance and self-care. This model emphasizes sustainable high performance through resilience, stress management, and contextual awareness, rather than pinpointing grit as a causal variable or tracing its effects via specific mediators. Therefore, the authors are recommended to revise their claim.

• I suggest the authors re-work the literature review and dedicate specific sections to life satisfaction, grit and life satisfaction, and the mediating role of SE. Also, it is highly recommended that they add another sub-section titled as theoretical background/framework.

• There is no need for 4 hypotheses. Only two hypothesis would be sufficient: one for the direct relationship between grit and life satisfaction and another for the mediating role of SE. Accordingly, the authors are suggested to rework the results and discussion section.

• More demographic information is needed. For example, the participants’ type of occupation is necessary. I suggest the authors provide more.

• The discussion of the findings should be more contextualized.

• Suggestions for further research is recommended to go beyond the limitations of the study.

• The theoretical implications should be presented persuasively.

Best regards,

Reviewer

Reviewer #2: 1. In the Introduction section why authors talking on system in UK and then shifting to UEA. Rather put in the system related to UAE employment and undergraduate.

2. Authors need to justify why only 242 response collected. Any method used like G*Power?

3. Why there is no pretest or pilot test conducted?

4. It would be better to provide a research framework in the literature part.

5. It would be better for readers to include why authors choose 'Grit" as it is more explored in the Western context. Provide some literature that this variable is less explored in Middle Eastern context.

**Do you want your identity to be public for this peer review?** For information about this choice, including consent withdrawal, please see our Privacy Policy

Reviewer #1: No

Reviewer #2: **Yes:** A. Devisakti

---

## [Author Response · Author response to Decision Letter 1]

7 Oct 2025

Dear Reviewers,

We sincerely thank you for your thoughtful and constructive feedback. Your comments and suggestions were insightful and have greatly contributed to strengthening the academic rigour of our manuscript. They also helped us to identify and correct several important points we may have previously overlooked.

Reviewer 1 Comments and Response

1. Abstract: the authors claim to measure ‘well-being’, which is incorrect.

Thank you for pointing this out. We have revised the abstract to accurately state that the study measured grit, self-efficacy and life satisfaction (Page 1).

2. Abstract: There is no need to refer to Beta and CI values.

We have removed the Beta and CI values from the abstract as suggested.

3. The opening paragraph of the introduction section is too broad.

Thank you for this helpful comment, which guided us in narrowing down the research focus more clearly. We have revised the introduction to better contextualise the experiences of employed university students residing in the UAE (Page 2).

4. The authors are suggested to more clearly explain the research problem and highlight significance of their study in the introduction section.

Thank you for this valuable suggestion. We have revised the introduction to more clearly outline the research problem and emphasise the significance of the study. These revisions have been incorporated in the second paragraph of the introduction section (Page 3).

5. The concept of life satisfaction requires further elaboration, as its current introduction lacks sufficient depth.

We have expanded the discussion of life satisfaction to provide a more detailed explanation. Specifically, we have added a description at the end of the paragraph titled “employed university students” in the literature review section (Page 4). In addition, we have elaborated on why life satisfaction was selected as a suitable measure of well-being towards the end of paragraph 1, under the subheading “theoretical framework” (Page 9).

6. The introduction and literature review sections should be clearly separated and numbered.

Thank you for this helpful suggestion. We have revised the manuscript to clearly separate the Introduction and Literature review sections. However, we did not number the sections to ensure compliance with the journal’s formatting guidelines (https://journals.plos.org/plosone/s/file?id=wjVg/PLOSOne_formatting_sample_main_body.pdf

)

7. I suggest authors avoid discussing measurement issues concerning grit in literature review.

Thank you for this constructive comment, as it helped us narrow the focus of the study to the most relevant information. Accordingly, we have removed the paragraph discussing measurement issues related to grit (Page 6).

8. The authors claim that the “OPAH model proposes that grit enhances life satisfaction via three psychological processes: needs satisfaction, emotional regulation and positive cognitions”. However, I think this is problematic. The OPAH model is about balancing optimal performance with optimal health by maintaining equilibrium between effort and recovery, performance and self-care. This model emphasizes sustainable high performance through resilience, stress management, and contextual awareness, rather than pinpointing grit as a causal variable or tracing its effects via specific mediators. Therefore, the authors are recommended to revise their claim.

We thank the reviewer for this insightful comment. We agree that the OPAH model primarily focuses on sustaining optimal performance by balancing effort with recovery. In the current study, however, our aim was to explore a potential mechanism through which grittier employed students may leverage the strength of grit to enhance life satisfaction.

The OPAH model of grit proposed by Datu et al (2021) suggests that grit can influence achievement and well-being (Paragraph , Page 9). Including this theory in the study was important, as it provides a theoretical rationale supporting the influence of grit on well-being (life satisfaction is a dimension of well-being). However, while the model highlights that grit may impact life satisfaction via multiple mediators, the proposed mediators are broad and somewhat vague. Therefore, we selected self-efficacy as a specific mediator because it represents a “positive cognition” and has been associated with several psychological benefits (“Self-efficacy” Paragraph 1, page 7). In Paragraph 2 (Page 9) we outline how self-efficacy could potentially mediated the relationship between grit and life satisfaction.

Furthermore, after carefully considering the theoretical rationale, we determined that including Fredrickson’s Broaden-and-Build theory was unnecessary, as focusing on self-efficacy within the OPAH framework provided a more logical and direct explanation for the research problem.

9. I suggest the authors re-work the literature review and dedicate specific sections to life satisfaction, grit and life satisfaction, and the mediating role of SE. Also, it is highly recommended that they add another sub-section titled as theoretical background/framework.

Thank you for this suggestion. In the literature review section, we have created separate sections for employed university students, grit and self-efficacy. The subsection of “employed university students” highlights the experiences unique to this group and provides a discussion on life satisfaction (Paragraph 1, Page 4). In addition, we have added a new subsection titled “Theoretical framework”, which explains the rationale for selecting self-efficacy as a mediator and describes how it may mediate between grit and life satisfaction (Paragraph 1 Page 9).

10. There is no need for 4 hypotheses. Only two hypothesis would be sufficient: one for the direct relationship between grit and life satisfaction and another for the mediating role of SE. Accordingly, the authors are suggested to rework the results and discussion section.

We have removed hypotheses H3 and H4, and revised the Discussion section accordingly (Paragraph 1, Page 19).

11. More demographic information is needed. For example, the participants’ type of occupation is necessary. I suggest the authors provide more.

Thank you for this helpful comment. While we did not collect data on participants’ specific occupations, we recorded the type of employment (full-time or part-time). Additionally, we collected other relevant demographic information, including marital status and whether participants have children, which has been included on the manuscript (Page 12), and discussed accordingly in the discussion section (Paragraph 3, page 20).

12. The discussion of the findings should be more contextualized.

Thank you for this valuable suggestion. We have revised the discussion section to better situate the findings within the current context (Paragraph 3, page 20).

13. Suggestions for further research is recommended to go beyond the limitations of the study.

Thank you for this helpful suggestion. We have revised the manuscript to provide recommendations for future research that extend beyond the limitations of the current study (Paragraph 1, Page 25; Paragraph 3, Page 26).

14. The theoretical implications should be presented persuasively.

We appreciate this constructive comment. We have revised the manuscript to present the theoretical implications more persuasively, drawing on the study findings and the underlying theoretical framework (Paragraph 3, Page 28).

Reviewer 2 Comments and Response

1. In the Introduction section why authors talking on system in UK and then shifting to UAE. Rather put in the system related to UAE employment and undergraduate.

Thank you for this helpful comment. We have removed details pertaining to employed university students in the UK and USA, and focused exclusively on the context of employed university students in the UAE (Paragraph 1, Page 2).

2. Authors need to justify why only 242 response collected. Any method used like G*Power?

Thank you for this comment. A G*Power analysis indicated that the ideal sample size for the study was 174. The final sample of 242 exceeds this requirement, ensuring that the sample size was sufficient for statistical analyses (Paragraph 3, Page 12).

3. Why there is no pretest or pilot test conducted?

We agree that conducting a pilot study could have further refined the study and enhanced its contextual relevance to the environment. However, the current study was conducted as part of the requirements for a master’s program. Due to time constraints associated with program completion, we were unable to conduct a pilot.

4. It would be better to provide a research framework in the literature part.

Thank you for this constructive suggestion. We have added a subsection in the literature review titled “theoretical framework”, which explains the relationships between grit, self-efficacy and life satisfaction among employed university students. This section also provides a rationale for the mediating role of self-efficacy in the relationship between grit and life satisfaction.

5. It would be better for readers to include why authors choose 'Grit" as it is more explored in the Western context. Provide some literature that this variable is less explored in Middle Eastern context.

Thank you for this valuable comment, which helped highlight the research problem and emphasise the novelty of the study. At the end of paragraph 1 in the introduction section (Page 2), we have indicated that although there is a growing interest in strengths-based approach in the Arab world, limited studies have adopted such an approach to study the experiences of employed university students residing in the UAE. This helps to establish that grit is a less explored strength-based construct in the UAE.

---

## [Editor Report · Decision Letter 1]

11 Feb 2026

Self-efficacy mediates the relationship between grit and life satisfaction in a sample of employed university students resident in the United Arab Emirates

PONE-D-25-33811R1

Dear Dr. Vally,

We’re pleased to inform you that your manuscript has been judged scientifically suitable for publication and will be formally accepted for publication once it meets all outstanding technical requirements.

Kind regards,

Frantisek Sudzina

Academic Editor

PLOS One
---

## [Editor Report · Acceptance letter]

PONE-D-25-33811R1

PLOS One

Dear Dr. Vally,

I'm pleased to inform you that your manuscript has been deemed suitable for publication in PLOS One. Congratulations! Your manuscript is now being handed over to our production team.

Kind regards,

on behalf of

Dr. Frantisek Sudzina

Academic Editor

PLOS One